# *Bacillus subtilis* Expressing Chicken NK-2 Peptide Enhances the Efficacy of EF-1α Vaccination in *Eimeria maxima*-Challenged Broiler Chickens

**DOI:** 10.3390/ani13081383

**Published:** 2023-04-18

**Authors:** Youngsub Lee, Inkyung Park, Samiru S. Wickramasuriya, Hyun S. Lillehoj

**Affiliations:** Animal Biosciences and Biotechnology Laboratory, United States Department of Agriculture, Agricultural Research Service, Beltsville, MD 20705, USA; youngsub.lee@usda.gov (Y.L.); inkyung.park@usda.gov (I.P.); samiru.sudharaka@usda.gov (S.S.W.)

**Keywords:** EF-1α vaccine, *Bacillus subtilis*, NK-lysin, coccidiosis, antimicrobial peptide, alternative to antibiotics

## Abstract

**Simple Summary:**

Despite the many advantages of recombinant vaccines against poultry coccidiosis, no commercialized subunit vaccines are available for coccidiosis prevention due to their limited efficacy. With a strict restriction of antibiotic uses for coccidiosis control in commercial poultry production, there is a timely need to explore novel strategies to enhance the efficacy of the recombinant vaccination strategy to reduce the economic costs due to avian coccidiosis. This study investigated novel ways to enhance the efficacy of the recombinant *Eimeria* elongation factor-1α (rEF-1α) recombinant vaccine against *Eimeria maxima* (*E. maxima*) infection in broiler chickens by co-administering the EF-1α vaccine with orally delivered *Bacillus subtilis* (*B. subtilis*) expressing chicken NK-2 peptide (cNK-2). This novel strategy resulted in enhanced host protection and resilience against *E. maxima* infection.

**Abstract:**

This study was conducted to investigate the synergistic effects of orally delivered *B. subtilis*-cNK-2 on vaccination with rEF-1α against *E. maxima* infection in broiler chickens. Chickens were assigned into the following five groups: control (CON, no *Eimeria* infection), non-immunized control (NC, PBS), component 1 (COM1, rEF-1α only), component 2 (COM2, rEF-1α plus *B. subtilis* empty vector), and component 3 (COM3, rEF-1α plus *B. subtilis*-NK-2). The first immunization was administered intramuscularly on day 4, and the second immunization was given one week later with the same concentration of components as the primary immunization. The immunization of *B. subtilis* spores (COM2 and COM3) was performed by oral administration given for 5 consecutive days a week later than the second immunization. On day 19, all the chickens except the CON group were orally challenged with *E. maxima* oocysts (1.0 × 10^4^/chicken). The results of the in vivo vaccination showed that all the chickens immunized with rEF-1α (COM1, COM2, and COM3) produced higher (*p* < 0.05) serum antibodies against EF-1α on 12 days post-*E. maxima* infection (dpi). The COM3 group showed a significantly (*p* < 0.05) higher average body weight gain (BWG) on 0–6, 6–9, and 0–12 dpi compared to those of the non-immunized chickens (NC). Immunization with rEF-1α alone (COM1) reduced the gut lesion score on 6 dpi and the fecal oocyst shedding on 9 dpi, whereas co-administration with *B. subtilis* spores (COM2 or COM3) led to further reduction in the lesion score. *E. maxima* infection increased the expression levels of IFN-γ and IL-17β in the jejunum, but these expressions were downregulated in the rEF-1α immunized (COM1) group and in the groups immunized with rEF-1α and orally treated with *B. subtilis* spores (COM2 or COM3) at 4 dpi. A reduced gene expression of occludin in the jejunum of the *E. maxima*-infected chickens on 4 dpi was upregulated following the immunization with COM2. Collectively, rEF-1α vaccination induced significant protection against *E. maxima* infection in the broiler chickens, and the efficacy of rEF-1α vaccination was further enhanced by co-administration with orally delivered *B. subtilis* spores expressing cNK-2.

## 1. Introduction

Coccidiosis is caused by several distinct species of apicomplexan parasites costing more than USD 14 billion in annual losses to the global poultry industry [1]. Inefficient feed utilization, impaired growth rate, and high mortality following parasite-induced intestinal damage are the major factors that are hindering optimum productivity [2,3]. Seven species of *Eimeria* have been identified in chickens; each species possesses different pathological potential [2]. Infection with specific *Eimeria* species is a predisposing risk factor for necrotic enteritis (NE) and can also result in alterations to the overall structure of the intestinal microbiota [4].

Due to the restriction of using antibiotics in poultry production, there has been increasing interest in developing alternative strategies to manage coccidiosis [5,6]. In commercial poultry production, coccidiosis is primarily managed through various conventional methods, including the use of synthetic drugs, ionophore antibiotics, and live attenuated coccidiosis vaccines [7,8,9]. However, the emergence of drug resistance to anticoccidial drugs and genetic variants of *Eimeria* strains, as well as the issue of residual virulence, have been consistently identified as potential risk factors [6,10,11]. Increasing scientific evidence indicates that recombinant anticoccidial vaccines are promising alternatives to antibiotics in the control of coccidiosis in poultry [12]. Over the last decades, recombinant protein technology has become efficient and relatively inexpensive through the use of microbial and other expression host systems [13]. In addition, recombinant protein vaccines are considered a safer approach than live parasite-derived vaccines due to their non-replicating nature and lack of infectious antigen components.

The immunodominant antigens of *Eimeria* trigger a specific immune response in the gut mucosa, which confers protective immunity against subsequent infections with the same or related *Eimeria* species in poultry [14,15,16]. Ideally, using a cross-reactive antigen of *Eimeria* parasites that can protect against species- and strain-variants of *Eimeria* parasites will lead to an effective coccidiosis control since *Eimeria* parasites are known to evolve quickly under environmental pressures [17]. EF-1α is an immunodominant protein identified as a shared antigen among *E. acervulina*, *E. tenella*, and *E. maxima* [18]. Lin et al. (2017) [16] demonstrated that EF-1α vaccination elicits cross-protective immunity against all *Eimeria* species and the immunization of young chickens with a cloned EF-1α gene from *E. tenella* elicited protective immunity against *E. maxima* infection in broiler chickens. Furthermore, we have recently demonstrated that EF-1α-induced protective immunity can further be enhanced by the co-administration with recombinant chicken IL-7 or chicken host antimicrobial peptide [19,20].

NK-lysin peptide 2 (cNK-2), an antimicrobial peptide produced by chicken cytotoxic T and natural killer (NK) cells, shows high specificity and cytotoxicity against all species of *Eimeria* parasites by disrupting the sporozoite membrane [21,22]. The characteristics and role of oral cNK-2 peptide treatment on coccidiosis in poultry have been reported previously [21,22,23]. Moreover, in a recent study, orally delivered cNK-2 peptide delivered by the *B. subtilis* spore expression system induced a significant protective effect against *E. acervulina* infection [24,25]. Compared to the chickens that received *B. subtilis* spores without cNK2 peptide, the chickens orally administered with *B. subtilis* expressing cNK-2 peptide showed significantly reduced fecal oocyst shedding, enhanced gut integrity, and higher innate immunity. Moreover, chickens receiving *B. subtilis*-cNK2 showed notably improved growth performance and rapid intestinal recovery. Therefore, in the present study, *B. subtilis*-EV and *B. subtilis*-cNK-2 were co-administered with the rEF-1α vaccine to assess their synergistic effects in broiler chickens experimentally infected with *E. maxima*.

## 2. Materials and Methods

### 2.1. Recombinant EF-1α and Recombinant B. subtilis Spores

The major antigen, recombinant EF-1α (GenBank Accession Number KX900609) expressed in *Escherichia coli* (BL21) was manufactured (GenScript, Inc., Piscataway, NJ, USA) [20]. Recombinant *B. subtilis* spores expressing empty vector (*B. subtilis*-EV) or *B. subtilis*-cNK-2 (RRQRSICKQLLKKLRQQLSDALQNNDD) were constructed and provided by US Biologic (Memphis, TN, USA) [22,24].

### 2.2. Animal Husbandry, Immunization, and E. maxima Challenge Infection

The schematic outline of the chicken experimental design is depicted in Figure 1. A total of 150 male day-old Ross 708 broiler chickens (30/group) were obtained from a local commercial hatchery (Longenecker’s Hatchery, Elizabethtown, PA, USA) and housed in Petersime brooder units. Each group was divided into five cages (5 cages/group) with feed and water available, ad libitum, throughout the experiment. The group information based on the vaccination is shown in Table 1 and includes the following groups: control (CON, no *Eimeria* infection), non-immunized control (NC, PBS), component 1 (COM1, 100 µg of rEF-1α), component 2 (COM2, 100 µg of rEF-1α plus 1 × 10^13^ cfu of *B. subtilis* empty vector), and component 3 (COM3, 100 µg of rEF-1α plus 1 × 10^13^ cfu of *B. subtilis*-NK-2). All immunizations except for the CON group were carried out intramuscularly (both sides of the thighs) with a mineral oil-based Montanide^TM^ ISA 78 VG adjuvant (ISA 78; Seppic, Fairfield, NJ, USA) in a 70/30 ratio. The first immunization of birds was administered on day 4 and secondary immunization was given with the same concentration of components as the primary immunization one week later. *B. subtilis*-EV and *B. subtilis*-cNk-2 in the COM2 and COM3 groups were orally administered for five days (from day 18 to day 22), respectively, with Montanide™ IMS 1313 N VGPR adjuvant (IMS 1313; Seppic, Puteaux, France) in a 50/50 ratio, as recommended by the adjuvant manufacturer.

All chickens except for the CON group were orally inoculated with freshly sporulated *E. maxima* strain 41A oocysts (1.0 × 10^4^/ chicken) through oral gavage. Individual body weight was recorded for all chickens on 0 (before *Eimeria* challenge infection), 6, 9, and 12 dpi, and then the average body weight gain per cage was calculated.

### 2.3. Sample Collection

Blood (12 chickens per group) was collected by heart puncture in serum gel tubes (S-Monovette) on day 20 post-secondary immunization. After blood collection, the chickens were humanely sacrificed by manual cervical dislocation as previously described [26]. The serum samples were separated by centrifugation at 2000× *g* for 10 min and subsequently kept at −20 °C until the antibody titer analysis was performed. The jejunum samples were collected at two time points of the experiment. On 4 dpi, the jejunum samples (6 chickens/group) were collected for gene expression analysis. A segment of intestinal jejunum tissues was collected and immediately placed in RNAlater. On 6 dpi, two equal 10 cm sections of the jejunum centering on Meckel’s diverticulum (6 chickens/group) were taken and the jejunal lesion score (0–4) was anonymously evaluated by four well-trained independent observers. Feces were collected from individual cages between 6 to 9 dpi, ground and homogenized with 3 L of water. Average oocyst production per cage was counted by three independent observers using a McMaster counting chamber. All procedures for sample collections were performed in the same manner as described in the previous study [20].

### 2.4. Indirect ELISA for Measurement of Anti-EF-1α Antibody Titer

The serum IgG antibody titers against rEF-1α were measured by an indirect ELISA as previously described [20]. In brief, rEF-1α antigen (1 μg/well) was coated into 96-well microtiter plates overnight at 4 °C. The plate was blocked with PBS containing 1.0% bovine serum albumin (BSA) for an hour at room temperature (R/T) and chicken serums (12 chickens/group) were applied, followed by incubation for 2 h at R/T on the plate shaker. The bounded antibodies against antigen were detected with avidin-horseradish peroxidase (HRP)-conjugated rabbit anti-chicken IgG secondary antibody (Sigma-Aldrich, St. Louis, MO, USA) diluted 1:5000 in PBS/0.1% BSA. The reaction was developed with TMB substrate (Sigma-Aldrich, St. Louis, MO, USA) and it was stopped by using 2 M sulfuric acid (50 μL/well), followed by optical density measurements at 450 nm (ELx-800, Biotek, Winooski, VT, USA). Each sample was analyzed in triplicates and the plate washings were carried out six times after every incubation step using a plate washer (ELx405, Biotek, Winooski, VT, USA) with PBS/T.

### 2.5. Quantitative RT-PCR Analysis

All the procedures for quantitative RT-PCR analysis, including RNA isolation and cDNA transcription, were performed in the same manner as described in the previous study [20,27]. Briefly, the jejunum samples were cut and opened longitudinally, and washed out three times with ice-cold Hank’s balanced salt solution (Sigma, St. Louis, MO, USA) to remove the gut contents and RNAlater. The mucosa layer was carefully scraped away using a surgical scalpel, and the collected mucosa was homogenized using a handheld homogenizer (TissueRuptor; Qiagen, Hilden, Germany). Total RNA was extracted using TRIzol reagent (Invitrogen, Carlsbad, CA, USA) followed by DNase digestion as described [25]. The purity and quantity of RNA was assessed using a NanoDrop spectrophotometer (Termo Fisher Scientifc, Model 2000, Waltham, MA, USA) at 260/280 nm and stored at −20 °C for further PCR amplification. Total RNA (5 μg) was reverse-transcribed to cDNA using a QuantiTect Reverse Transcription Kit (Qiagen, Hilden, Germany) according to the manufacturer’s instructions. The cDNA was analyzed using SYBR Green qPCR Master Mix (PowerTrack, Applied Biosystems, Vilnius, Lithuania) in triplicates using Applied Biosystems QuantStudio 3 Real-Time PCR Systems (Life Technologies, Carlsbad, CA, USA). The thermocycling program was as follows: denaturation at 95 °C for 10 min, followed by amplification at 58 °C for 1 min for 40 cycles. The gene markers included two proinflammatory cytokines. IFN-γ (GenBank accession no. NM_205149; F: 5-AGCTGACGGTGGACCTATTATT-3′ and R: 5′-GGCTTTGCGCTGGATTC-3), IL-17F (GenBank accession no. JQ776598; F: 5′-TGAAGACTGCCTGAACCA-3′ and R: 5′-AGAGACCGATTCCTGATGT-3′), and TJ protein Occludin (GenBank accession no. NM_205128; F: 5′-GAGCCCAGACTACCAAAGCAA-3′ and R: 5′-GCTTGATGTGGAAGAGCTTGTTG-3′) were selected for gene expression analysis. The housekeeping gene β-actin (GenBank accession no. NM_205518; F: 5′-CACAGATCATGTTTGAGACCTT-3′, and R: CATCACAATACCAGTGGTACG) was used as a reference gene for normalization of target gene expression. Each analysis was performed in triplicate and individually normalized transcripts to those of β-actin were quantified using a 2^−ΔΔCt^ method [28].

### 2.6. Statistical Analysis

Statistical analyses were performed using the SPSS 20.0 statistical software (SPSS Inc., Chicago, IL, USA) for Windows. One-way analysis of variance (ANOVA) with Tukey’s multiple comparison test was used and the difference between groups was considered statistically significant when *p* values were less than 0.05.

## 3. Results and Discussion

The main objective of this study was to develop a novel strategy to improve the efficacy of rEF-1α vaccination against *Eimeria* infection in poultry by simultaneous injection with chicken NK-lysin peptide. NK-lysin peptide was delivered using transgenic *B. subtilis* spores carrying cNK-2 peptide. In order to evaluate the synergistic effect of rEF-1α and cNK-2, we conducted various analyses including BWG change, serum antibody titer to rEF-1α, lesion score, fecal oocyst shedding, and expression changes of immune-related genes in the jejunum, all of which have been used as parameters to measure protective effects against *E. maxima* infection.

### 3.1. Body Weight

No significant changes in body weight due to the immunization were observed before the infection, and the average body weights before infection were as follows: CON (705.7), NC (692.3), COM1 (678.1), COM2 (678.2), and COM3 (687.1) (Table 2). Following the *E. maxima* infection, the chickens in the infected groups (NC, COM1, COM2, and COM3) showed significantly (*p* < 0.05) lower BWG than the unchallenged chickens (CON) during 0–6, 6–9, and 0–12 dpi. However, overall, the chickens immunized with rEF-1α (COM1 and COM2) showed a greater BWG compared to the unimmunized chickens (NC). Notably, the chickens co-immunized with rEF-1α and *B. subtilis*-NK-2 (COM3) showed a significantly higher BWG compared to those of the chickens in the other groups (NC, COM1, and COM2) during 0–6 (*p* < 0.01), 6–9 (*p* < 0.05), and 0–12 dpi (*p* < 0.01). There was no statistical difference in BWG between the groups at 9–12 dpi. The most severe body weight loss occurred during 0–6 dpi. However, weight loss was less severe during 6–9 dpi and a high rate of body weight recovery was observed between 9–12 dpi. Nevertheless, significant body weight changes were detected between the uninfected (CON) and infected chickens (NC), but all the chickens recovered after 7 dpi. Interestingly, body weight changes observed in this study following the *E. maxima* infection were consistent with previous in vivo studies in poultry, showing recovery after 7 days [29,30,31].

### 3.2. Serum Antibody Titer

As shown in Figure 2, the chickens infected with *E. maxima* (NC, COM1, COM2, and COM3) exhibited higher serum antibody titers to rEF-1α when compared to those of the uninfected chickens. However, among the chickens infected with *E. maxima*, the chickens immunized with rEF-1α (COM1, COM2, and COM3) showed greater (*p* < 0.05) antibody titters than the non-immunized chickens (NC). Significant statistical differences were not found among the immunized groups (COM1, COM2, and COM3). Recent studies [32,33] have demonstrated that higher levels of antibodies correlated with higher levels of neutralizing antibodies. Therefore, we expected that the coadministration of *B. subtilis* strains would contribute to the improved antibody production against the rEF-1α antigen. However, co-injection of the rEF-1α vaccine with cNK lysin (COM1, COM2, and COM3) did not enhance the serum antibody titers.

### 3.3. Lesion Score and Fecal Oocyst Shedding

*E. maxima* infection induced gut lesions (*p* < 0.01) in the jejunum of the chickens at 7 dpi (Figure 3A). Among the chickens infected with *E. maxima*, jejunums of the chickens in the unimmunized group (NC) showed the highest gut lesion score. The chickens immunized with rEF-1α alone (COM1) showed reduced lesion scores which were not statistically significant compared with that of the chickens in the NC group. The chickens communized with rEF-1α and *B. subtilis*-cNK2 spores (COM2 and COM3) showed a significantly (*p* < 0.05) lower gut lesion scores than the chickens in the NC group. The average lesion scores of the COM2 and COM3 groups were 1.7 and 1.8, respectively, and there was no significant difference between these groups. Fecal oocyst shedding showed a similar graph pattern with a jejunal lesion score (Figure 3B). As expected, no oocyst was found in the uninfected group (CON). The chickens in the NC group showed the highest oocyst production whereas the chickens immunized with rEF-1α (COM1 and COM2) showed a significantly (*p* < 0.05) reduced fecal oocyst production compared to the NC group. Interestingly, the chickens immunized with rEF-1α and *B. subtilis*-cNK-2 (COM3) showed a 28% reduction (*p* < 0.01) in the fecal oocyst output in comparison to the NC group. However, the chickens immunized with rEF-1α and *B. subtilis*-EV (COM2) did not exhibit a reduction in the fecal oocyst count; these findings indicate the immune-enhancing effects of NK-lysin peptide delivered in *B. subtilis*-cNK-2 spores [20,21,22,24].

### 3.4. Gene Expression Level Changes in the Jejunum

*E. maxima* infection significantly (*p* < 0.01) upregulated the expression level of the proinflammatory cytokines (IFN-γ and IL-17β) (Figure 4A,B). In comparison with the NC group, the chickens immunized with COM3 showed a significantly (*p* < 0.05) down-regulated mRNA expression level of IFN-γ. Furthermore, the mRNA expression level of IL-17β was significantly (*p* < 0.05) decreased in the chickens immunized with COM1 and COM2 compared to that of the chickens in the NC group. The chickens immunized with COM3 also showed a reduced expression level of IL-17β but, no statistical difference was found because of the wide deviations between samples. Proinflammatory cytokines such as IL-1, IL-8, and IFN-γ are involved in early inflammatory responses and play an important role in controlling intracellular parasites and their elimination by perpetuating the proinflammatory responses [34,35,36]. As shown in a previous study [37], the expression of proinflammatory cytokines (IFN-α, IL-1β, IL-6, and IL-17) gradually increased, peaked at 4–6 dpi, but rapidly decreased after 7 dpi post-*E. maxima* infection (1 × 10^4^/chicken). These patterns of the mRNA expressions imply that the host recovers from *Eimeria* infection after 7 dpi, and this result is consistent with the BWG change following *E. maxima* infection shown in this study (Table 2). More importantly, the chickens infected with a two-fold higher oocyst dose of *E. maxima* as a secondary infection showed a decreased expression level of pro-inflammatory cytokines (IL-1β, IL-6 and IL-17) compared to that of the primary infection. These results provide evidence that the adaptive immunity activated by primary infection or vaccination protects the gut with less damage after *Eimeria* challenge infection. Accordingly, the relatively down-regulated expression level of IFN-γ and IL-17β in the vaccinated groups (COM1, COM2 and COM3) compared with the unimmunized group (NC) indicate enhanced disease resistance against *Eimeria* infection in the vaccinated groups.

Following the *E. maxima* challenge infection, significant intestinal damage occured evidenced by the reduced mRNA expression level of Occludin (*p* < 0.05) (Figure 3C). Among the chickens immunized with rEF-1α, the COM2 group showed a significant (*p* < 0.05) increase in the gene expression level of Occludin compared to the NC group. Notably, the Oocludin mRNA expression level of the COM2 group increased to that of the uninfected chickens (CON). The TJ proteins play a role in controlling cell proliferation and differentiation, which are critical processes for the formation and maintenance of tissue barriers [38]. Hence, the upregulated mRNA expression level of Occludin in the immunized groups (COM1, COM2, and COM3), compared to the NC group implies strengthened gut barrier function induced by vaccination with rEF-1α and *B. subtilis* spores (Figure 4C).

## 4. Conclusions

Recombinant protein vaccines are comparatively cost-effective, scalable in mass production, and, unlike antibiotics, do not leave any residual side effects. In addition, *B. subtilis* is widely acknowledged for its high resistance to environmental stresses, and its stability and safety make it uniquely qualified as an environmental bacterium for the oral delivery of peptides [39]. In this study, the rEF-1α vaccination of young broiler chickens induced a partial protective immunity against *E. maxima* infection. Furthermore, the co-administration of the recombinant EF-1α vaccine with *B. subtilis* expressing cNK-2 antimicrobial peptide greatly improved the protection against the *E. maxima* challenge infection and provided beneficial effects in the host protective immune response to coccidiosis. Our results provide novel evidence that the efficacy of the recombinant vaccine against *E. maxima* infection can be enhanced by the simultaneous injection with coccidia-specific antimicrobial peptide in young chickens. We expect our findings to provide a good foundation for the advancement of next-generation coccidiosis vaccines and other parasite vaccines.

## Figures and Tables

**Figure 1 animals-13-01383-f001:**
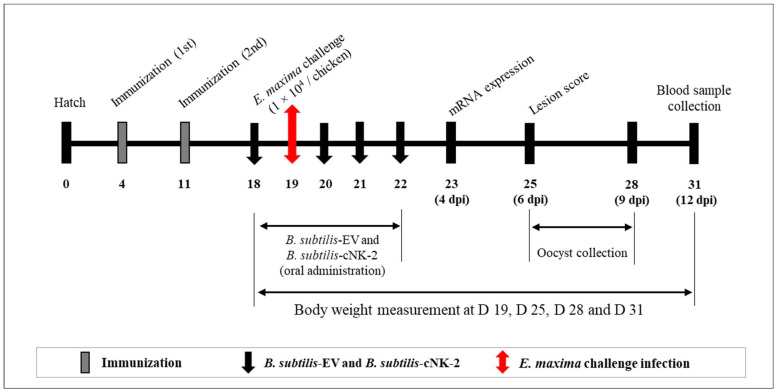
Schematic representation of the experimental designs and treatment information. Chickens except the CON and NC groups were immunized on day 4 and 11 with EF-1α, and orally co-immunized with either *B. subtilis*-cNK-EV or *B. subtilis*-cNK-2 from day 18 to 22. On day 19, all chickens except the CON group were orally challenge infected with sporulated-*E. maxima* (1 × 10^4^/chicken). Abbreviations: EV, *B. subtilis* (empty vector); cNK-2, *B. subtilis* expressing cNK-2; DPI, days post-*E. maxima* infection.

**Figure 2 animals-13-01383-f002:**
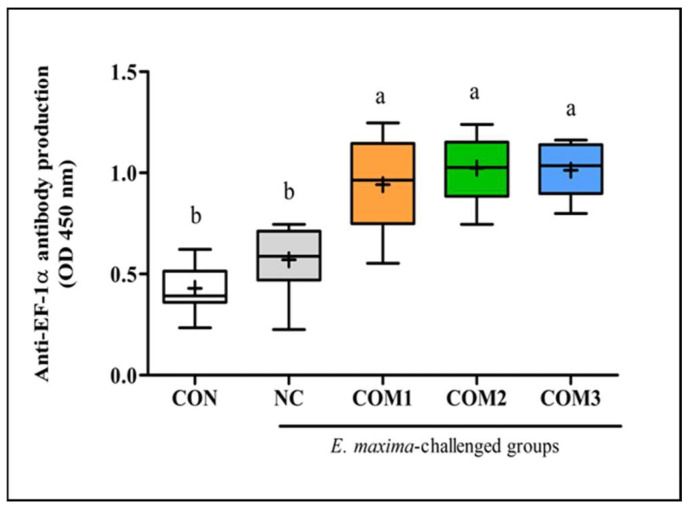
Synergetic effect of recombinant EF-1α and *B. subtilis* expressing cNK-2 peptide on anti-EF-1α antibody titer post-*E. maxima* infection. a,b different letters between group denote significant differences (*p* < 0.05) according to the Tukey’s multiple range test (*n* = 30). Abbreviations: CON, non-immunized and non-infected control; NC, non-immunized and infected control; COM, component; nd, not detected.

**Figure 3 animals-13-01383-f003:**
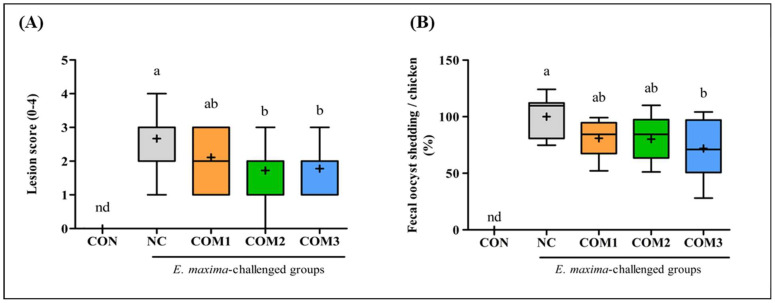
Synergetic effect of recombinant EF-1α and *B. subtilis* expressing cNK-2 peptide on the jejunal lesion score (**A**), and fecal oocyst shedding (**B**) post-*E. maxima* infection. a,b different letters between group denote significant differences (*p* < 0.05) according to the Tukey’s multiple range test (*n* = 30). Abbreviations: CON, non-immunized and non-infected control; NC, non-immunized and infected control; COM, component; nd, not detected.

**Figure 4 animals-13-01383-f004:**
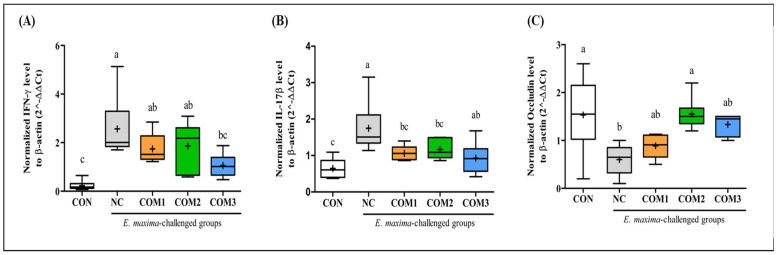
Synergetic effect of recombinant EF-1α and *B. subtilis* expressing cNK-2 peptide on intestinal expression of IFN-γ (**A**), IL-17β (**B**) and Occludin (**C**) post-*E. maxima* infection. a–c different letters between group denote significant differences (*p* < 0.05) according to the Tukey’s multiple range test (*n* = 30). Abbreviations: CON, non-immunized and non-infected control; NC, non-immunized and infected control; COM, component; nd, not detected.

**Table 1 animals-13-01383-t001:** Group information, immunization and *Eimeria* challenge infection status.

Treatment	Abbreviation	Description	rEF-1α	*B. subtilis*-EV	*B. subtilis*-cNK-2	*E. maxima* Challenge
Non-infected group	CON	PBS/ISA 78 VG	–	–	–	–
Infected group	NC	PBS/ISA 78 VG	–	–	–	1 × 10^4^/birds
EF-1α alone	COM1	EF-1α/ISA 78 VG	100 µg	–	–	1 × 10^4^/birds
EF-1α/*B. subtilis*-EV	COM2	EF-1α/ISA 78 VG	100 µg	10^13^ cfu/mL	–	1 × 10^4^/birds
EF-1α/*B. subtilis*-cNK-2	COM3	EF-1α/ISA 78 VG	100 µg	–	10^13^ cfu/mL	1 × 10^4^/birds

Abbreviations: CON, non-immunized and non-infected control; NC, non-immunized and infected control; COM, component; SD, standard deviation; DPI, days post-*E. maxima* infection; BW, body weight; BWG, body weight gain.

**Table 2 animals-13-01383-t002:** Effect of *B. subtilis*-EV or *B. subtilis*-cNK-2 on rEF-1α vaccination based on body weight gain post-*E. maxima*-challenge infection.

Treatment	CON	NC	COM1	COM2	COM3	SD (±)	*p*-Value
BW (g)							
Initial BW	77.2	77.5	77.1	77.1	77.6	0.62	0.84
Before infection	705.7	692.3	680.4	678.2	687.1	30.53	0.352
BWG (g)							
0–6 DPI	249.0 ^a^	132.8 ^c^	167.4 ^bc^	165.0 ^bc^	181.2 ^b^	50.3	*p <* 0.01
6–9 DPI	420.1 ^a^	338.7 ^b^	360.5 ^ab^	375.9 ^ab^	412.2 ^a^	50.34	*p <* 0.05
9–12 DPI	399.6	357.0	375.8	374.3	389.9	32.29	0.19
0–12 DPI	1068.6 ^a^	828.5 ^d^	903.7 ^cd^	915.3 ^bc^	983.3 ^b^	92.65	*p* < 0.01

The results were estimated using one-way ANOVA followed by Tukey’s test (*n* = 30). Values within column bearing different lowercase letters as superscripts differ significantly at *p* < 0.01 or *p* < 0.05. Abbreviations: CON, non-immunized and non-infected control; NC, non-immunized and infected control; COM, component; SD, standard deviation; DPI, days post-*E. maxima* infection; BW, body weight; BWG, body weight gain.

## Data Availability

The data presented in this study are available on request from the corresponding author.

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
