# Peer review of "Bacillus subtilis Expressing Chicken NK-2 Peptide Enhances the Efficacy of EF-1α Vaccination in Eimeria maxima-Challenged Broiler Chickens"

_animals, 2023, doi:10.3390/ani13081383_

Round 1
Reviewer 1 Report
Comments for the authors
Major comments
Introduction
- Add a paragraph on the economic impact of E. maxima infection and the cost-benefit of vaccination
Materials and Methods
- L116-120: provide details about the euthanasia of chickens experimental animals (e.g., mean BW, age, parity distribution, vaccinations)
Discussion
- You could underline the significance and economic impact of your results and the benefits of vaccination in comparison to antibiotics use
Minor comments
- L46: .. $ 14 billion in annual losses..
- L87: ad libitum (italics)
- L157:.. Results and Discussion
- L167: .. to immunization were observed before
- L182: .. in vivo (italics)
Author Response
Reviewer’s Comments and Author’s Responses
# Reviewer 1, comments for the authors
Major comments
Comment 1. Introduction
- Add a paragraph on the economic impact of E. maxima infection and the cost-benefit of vaccination
A: Thank you for your response. We have added the significance and economic benefits of recombinant protein vaccine in introduction section. Please see the line 50-61 highlighted in yellow.
Comment 2. Materials and Methods
- L116-120: provide details about the euthanasia of chickens experimental animals (e.g., mean BW, age, parity distribution, vaccinations).
A: Thank you for your response. In accordance with the reviewer’s suggestion, we have included a description of the method used to sacrifice chickens in the materials and methods sections. Please see the line128-132 highlighted in yellow.
Comment 3. Discussion
- You could underline the significance and economic impact of your results and the benefits of vaccination in comparison to antibiotics use.
A: Thank you for your response. We have further discussed the significance and economic benefits of recombinant protein vaccine in conclusion section. Please see the line 295-298 highlighted in yellow.
Minor comments
Comment 1. L46: .. $ 14 billion in annual losses..
A: Thank you for your response. We have revised sentence (line 46) according to reviewer’s comment.
Comment 2. L87: ad libitum (italics)
A: Thank you for your response. We have revised a typo (line 99) according to reviewer’s comment.
Comment 3. Results and Discussion
A: Thank you for your response. We have revised a typo (line 174) according to reviewer’s comment.
Comment 4. to immunization were observed before
A: Thank you for your response. We have revised a typo (line 184) according to reviewer’s comment.
Comment 5. L182: .. in vivo (italics)
A: Thank you for your response. We have revised a typo (line 199) according to reviewer’s comment.

Reviewer 2 Report
This is a well-written manuscript reporting the effect of recombinant vaccines on growth performance, immunity, and gut health in broilers challenged with Eimeria maxima. This manuscript addressed an important gut health issue in broilers.
There are minor comments.
1. The tables and figures should stand alone. All abbreviations should be defined, and N=??.
2. Oocyst shedding and gut permeability data could strengthen the results but are not essential. The body weight and lesion score can be essential parameters for this study.
Author Response
Reviewer’s Comments and Author’s Responses
# Reviewer 2 comments for the authors
Comments and Suggestions for Authors
This is a well-written manuscript reporting the effect of recombinant vaccines on growth performance, immunity, and gut health in broilers challenged with Eimeria maxima. This manuscript addressed an important gut health issue in broilers.
There are minor comments.
- The tables and figures should stand alone. All abbreviations should be defined, and N=??.
A: Thank you for your comment. We have revised footnotes including abbreviations under the tables and figures.
- Oocyst shedding and gut permeability data could strengthen the results but are not essential. The body weight and lesion score can be essential parameters for this study.
A: Thank you for your constructive comments and for spending time reviewing this article.
